# Giant Trees Exhibited Great Cooling Effect in Residential Area Southwest of China

**Rongfei Zhang** [1,2,3,*] **and Ziyan Zhao** [4]

1 College of Environment and Ecology, Chongqing University, Chongqing 400045, China
2 Key Laboratory of the Three Gorges Reservoir Region's Eco-environment, Ministry of Education, Chongqing University, Chongqing 400045, China
3 Guangdong Institute of Eco-Environment & Soil Science, Guangzhou 510650, China
4 School of Business Administration, Chongqing Technology and Business University, Chongqing 400067, China
* Correspondence: rongfei330@cqu.edu.cn

**Abstract:** In recent years, extreme weather has become more and more frequent. The extreme heat in summer is a serious threat to human health. Chongqing is one of the hottest cities in China, and the high temperature in summer can cause skin cancer and heat stroke. Green plants in residential areas play a great role in cooling down air temperature. However, there are no accurate numerical references for which plants have a better cooling effect. Therefore, seven species of trees with the highest planting rate were selected in seven residential areas as research objects in this study. By measuring the temperature under the trees and related control factors, this study was conducted to clarify the following: (1) Which species of tree has the best cooling effect? (2) Whether tree species and size matter with the cooling effect. (3) What are the controlling factors and principles that affect the tree cooling effect? The results showed that: (1) Giant trees have the best cooling effect. (2) The cooling effects of *Ficus virens*, *Camphor* tree, and *Ficus macrocarpa* were significantly better than the other four species of trees. (3) The high rate of water replenishment in plants gives giant trees a greater cooling effect. This study is of great practical significance to the selection of plants in the green belt of residential communities, and has great theoretical significance to understand the principle of the plant cooling effect.

**Keywords:** cooling effect; giant trees; temperature difference; deep soil; extreme heat

## 1. Introduction

In recent years, the frequency of extreme weather has been increasing, especially extreme heat weather, which poses a serious threat to people's health [1,2]. Extreme heat can cause serious direct damage to the human body, such as skin cancer and heat stroke [3], and it also can make urban roads very hot and potentially threaten the tires of vehicles [4]. With the rapid development of cities, the proportion of concrete buildings and pavement is increasing, which leads to an increasingly serious urban heat island effect [5–7]. To mitigate the heat island effect, urban green belts are widely used. Urban green belts, occupying a very important position in the city, have the effect of purifying the environment and beautifying the city. It is easy to understand that the plant green belt in residential areas can improve resistance to high temperatures to some extent [8]. The green belt has been extensively studied [9–13]. For example, Amati and Yokohari argued that a new green space planning concept should be implemented that explicitly refers to the green belt's role in restoring landscapes. [9]. Boentje and Blinnikov found that a trend toward less tree cover and more suburban development in the immediate vicinity of a large city is likely to result in worsening air quality and negative impacts on wildlife and opportunities [10]. Kowarik indicated that the new greenway may reduce environmental inequity in Berlin as it largely intersects neighborhoods where disadvantaged status coincides with poor access

to urban green space [11]. Pathak et al. evaluated the performance of some tree species for green belt development to mitigate traffic-generated noise, then the most suitable plant species for green belt development in urban areas were identified and recommended [12]. Prajapati and Tripathi evaluated the Air Pollution Tolerance Index (APTI) of the green belt by analyzing important biochemical parameters, then identified and recommended the most suitable plant species for green belt development in urban areas for long-term air pollution management [13]. Kubilay concluded that mitigation solutions for the urban heat island effect and heat waves, which should include vegetation, evaporative cooling pavements, and neighborhood morphology, are assessed in terms of pedestrian comfort and building energy consumption [14]. Manickathan found that the shading provided by trees improves thermal comfort more than that of the transportive cooling effect of a single row of trees, and taller trees are more beneficial because the vegetation canopy with high leaf temperatures is further away from the pedestrian level [15]. It can be concluded that the green belt can be composed of different plants, such as various grass, shrubs, and arbor trees, and the functions of each plant show obvious differences. Obviously, the green belt has a cooling effect on air temperature; therefore, a variety of plants are planted near residential buildings to relieve the heat [16]. There is a Chinese proverb, it is good to enjoy the shade under giant trees. It is a good place under the giant tree where the temperature is cooler, which helps people cope with the hot summer more easily. Trees are well known for shading, and their transpiration has a significant cooling effect [17]. However, the specific cooling principle and process are often overlooked in the cooling process of trees.

Chongqing is one of the hottest cities in China, with a subtropical humid monsoon climate [18]. It is located on the southeastern edge of the Sichuan Basin in southwest China, among several parallel mountain ranges that run in the same direction [19]. This terrain is called parallel Ridge Valley and its geographical environment characteristics have a great influence on climate. The southeast monsoon from the Pacific Ocean and the southwest monsoon from the Indian Ocean are blocked by parallel mountains, such as tall mountains around the basin, making it difficult to cross these natural barriers and send cold air to Chongqing inside the basin [20]. Even if there is a chance to break through the siege, when cold air sinks down into the basin from the tall mountains on all sides, it absorbs the surrounding heat, causing water vapor to evaporate and the air temperature to rise, forming the dry hot foehn wind, which exacerbates the hot weather in Chongqing. Chongqing's urban parallel low-lying valley leads to high air density at the bottom of the atmosphere. A dense atmosphere during the day in summer has little effect on weakening solar radiation, while thick clouds at night prevent ground heat radiation into the air [21]. In addition, high-rise and dense buildings block the air circulation, weakening the wind speed, and the hot air cannot diffuse in Chongqing city [22]. In densely populated urban areas, and developed industrial and commercial areas, there is a lot of equipment with high energy consumption, and more heat is discharged into the air, which further intensifies the heat island effect in the Chongqing urban area [23].

Therefore, the temperature under different tree species was studied in seven residential communities in the Nanan district of Chongqing in order to clarify the related control factors. The objectives of the paper are: (1) Which species of tree has the best cooling effect? (2) Whether tree species and size matter with the cooling effect. (3) What are the controlling factors and principles that affect the tree cooling effect? This study is of great practical significance to the selection of plants in the green belt of residential communities and has great theoretical significance to understand the principle of the plant cooling effect.

## 2. Materials and Methods

### 2.1. Study Area

The study area was situated in the Nanan district (106°3′14″–106°47′2″ E, 29°27′2″–29°37′2″ N) of Chongqing, China (Figure 1). The area is located in the eastern Sichuan parallel range-valley area, with anticline and syncline distributed in parallel, constituting the combination of low mountains, hills, small flats, and rivers [24,25]. The elevation of

Chongqing's main urban area is mostly between 168 and 400 meters. The highest peak of the city is Yintiaoling, the main peak of Jieliang Mountain on the eastern edge of Wuxi County, 2796.8 meters above sea level. The lowest location is the water surface of the Yangtze River in Wushan County, 73.1 meters above sea level. The elevation difference in Chongqing is 2723.7 meters [26]. Chongqing has a subtropical monsoon humid climate, the average annual temperature is 16~18 °C, the Yangtze River valley Banan, Qijiang, Yunyang, and other places are up to 18.5 °C; the southeast Qianjiang, Youyang, and other places are 14–16 °C; the northeast higher elevation of Chengkou is only 13.7 °C; and the hottest month average temperature is 26–29 °C and the coldest month average temperature is 4–8 °C. The annual average relative humidity in Chongqing is mostly 70%–80%, which is a high humidity area in China [27]. The annual sunshine duration is 1000–1400 hours, and the sunshine percentage is only 25%–35%. With a subtropical monsoon climate, the Nanan district is rich in heat, abundant rainfall, long frost-free period, and little wind [19]. The annual average temperature is 18.5 °C and rainfall is approximately 1097.8 mm [28]. In our study, 7 residential communities were selected, which were built around the year 2000.

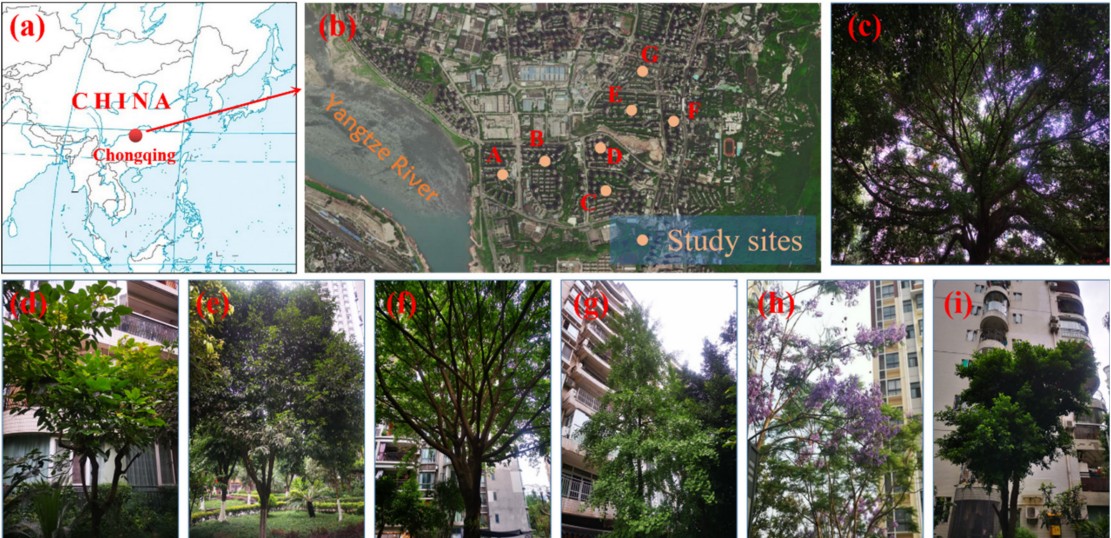

**Figure 1.** Study location and sites. Note that (**a**–**g**) are residential communities; (**c**–**i**) are *Ficus macrocarpa*, *Magnolia* tree, *Osmanthus fragrans*, *Ficus virens*, *Ginkgo biloba*, *Jacaranda* tree, and *Camphor* tree, respectively.

*2.2. Experiment Design and Field Measurements*

In our study, the most common 7 species of trees were selected in 7 residential communities, and for each species, 6 trees were selected based on different DBH (Diameter at breast height). All 6 trees of the same species were numbered from 1 to 6 (Control No.) according to DBH from large to small to make it easier to distinguish. The data were recorded from 1 June 2021 to 31 September 2021. As shown in Figure 2, the typical hottest days in each of the four months were selected as experimental days, which are 14 June, 14 July, 4 August, and 25 September, respectively.

To obtain deep soil temperature, soil temperature sensors (RR-7710, RS-Ltd., Hong Kong, China) were installed at 60 cm depth, the data was automatic continuous monitoring once every 5 minutes. Air temperature sensors (10TH) were used to measure air temperature automatically and continuously once every 5 minutes at 1.5 m height. *TDP* (Thermal Dissipation Probes, automatic continuous monitoring once every 5 minutes) were implanted in tree trunks at 1.5 m height to monitor the temperature of sap flow. The leaf area index (LAI) was measured once every 10–15 days using an LAI-2200C (LI-COR Ltd., Lincoln, NE, USA). The study was just in the peak growth period, and LAI did not change much; therefore, the data of LAI measured on 1 August were used in the study. Canopy height and canopy area were estimated by unmanned aerial vehicle (UAV, Dji-M600Pro) photos on 1 August. In

addition, all these 7 types of trees were near to each other (<1 km), and solar radiation and wind conditions were almost the same for them. Therefore, these factors were not the controls for the difference in cooling effect among different trees.

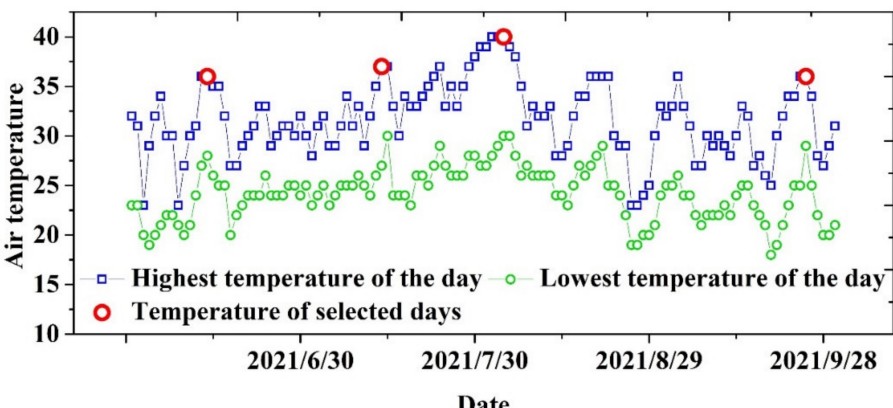

**Figure 2.** Air temperature for study sites during 1 June and 31 September 2021.

*2.3. Multiple Linear Regression*

To clarify the relationship between DTa (temperature difference between the tree trunk and deep soil) and its control factors, multiple linear regression was used in this study. Multiple linear regression is a generalization of simple linear regression, which studies the quantitative dependence between a dependent variable and multiple independent variables. The mathematical formula of multiple linear regression can be expressed as [29]:

$$y = \beta_0 + \beta_1 x_1 + \cdots + \beta_p x_p + \varepsilon \tag{1}$$

where $y$ is the dependent variable, and it is the random quantitative observation value. $\beta_0$ is the constant term, $x_1, \ldots, x_p$ are independent variables, $\beta_1, \ldots, \beta_p$ are partial regression coefficients. $B_i$ ($i$ = 1, 2, 3, $\ldots$, $p$) denotes when other independent variables are fixed, each unit change in the independent variable alone causes the average change in the dependent variable $y$. $\varepsilon$ is the random error, also known as residual, which is the part of the change in y that cannot be explained by independent variables and is subject to the distribution of $N$ (0, $\sigma^2$).

The multiple linear regression equation estimated from the sample is:

$$\hat{y} = b_0 + b_1 x_1 + \cdots + b_p x_p \tag{2}$$

where $\hat{y}$ is the average estimated value or average predicted value of the dependent variable y when a set of fixed values are taken for each $x$; and $b_0, \ldots, b_x$ are the sample estimate values of $\beta_1, \ldots, \beta_p$.

## 3. Results

*3.1. Comparison of Air Temperature under Different Trees*

As shown in Figure 3a–d, for all tree species, the air temperature under the trees with larger DBH is lower than that with smaller DBH. The air temperature of the largest DBH in June is 33.8 °C, 33.8 °C, 34.8 °C, 34.4 °C, 34.3 °C, 34.7 °C, and 33.4 °C for *Ficus virens*, *Camphor* tree, *Ginkgo biloba*, *Osmanthus fragrans*, *Magnolia* tree, *Jacaranda* tree, and *Ficus macrocarpa*, respectively. The air temperature under different trees also exhibited obvious differences among species. For example, the air temperature of species 6, i.e., *Jacaranda* tree, is the highest among all species from June (Mean = 35.4 °C) to September (Mean = 35.5 °C). The lowest temperature tree species is *Ficus virens* (Mean = 34.5 °C) both in June and September. In addition, the cooling effect of different tree species also showed seasonality. For example, the air temperature under *Ficus macrocarpa* is higher than *Ficus virens* in June

(Mean = 34.8 °C) and September (Mean = 34.9 °C), but it is lower in July (Mean = 35.6 °C) and August (Mean = 37.8 °C) than *Ficus virens* (Mean = 36.1 °C and 38.4 °C, respectively).

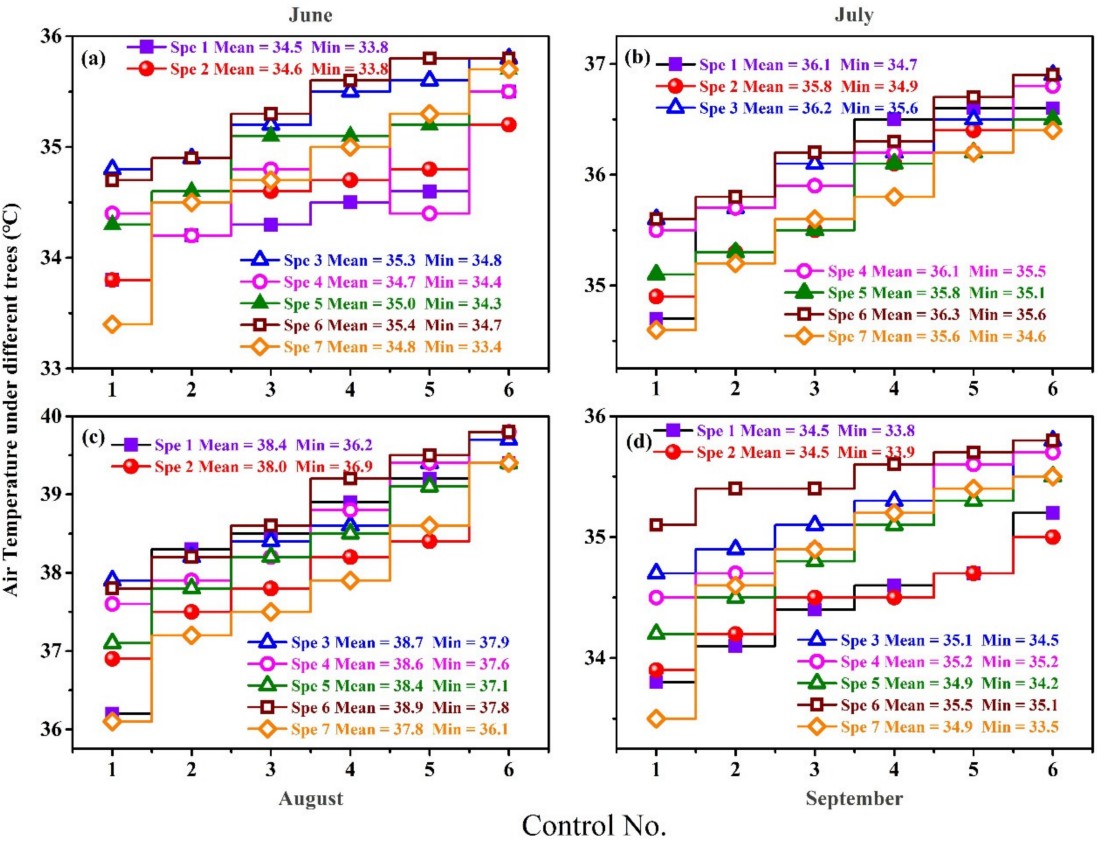

**Figure 3.** Measured air temperature under different trees in different month, and (**a**–**d**) are for June, July, August and September, respectively. Note that Spe 1–7 are *Ficus virens*, *Camphor* tree, *Ginkgo biloba*, *Osmanthus fragrans*, *Magnolia* tree, *Jacaranda* tree, and *Ficus macrocarpa*, respectively. Six trees of each species were distinguished using control No. 1–6 based on DBH size from large to small.

### 3.2. Impact of Temperature Difference between the Tree Trunk and Deep Soil

As shown in Figure 4, the temperature under trees and the temperature difference between the tree trunk and deep soil exhibited a linear positive correlation for all species ($R^2 > 0.27$, $P < 0.01$). The relationship persisted when all the trees were considered as a whole ($R^2 \geq 0.52$). Their correlations show some seasonality, with July ($R^2 = 0.79$) and August ($R^2 = 0.71$) showing higher correlations than that in June ($R^2 = 0.52$) and September ($R^2 \geq 0.64$). The temperature difference between the tree trunk and deep soil is linearly negatively correlated with DBH, which means that the temperature difference in the tree with the largest DBH is the lowest. As shown in Figure 5a,b, the temperature difference between the tree trunk and deep soil exhibited obvious differences among species. The temperature difference in *Ficus virens* (Mean = 2.62 ± 0.27 °C), *Camphor* tree (Mean = 2.40 ± 0.24 °C), and *Ficus macrocarpa* (Mean = 2.38 ± 0.12 °C) is lowest, and it is highest for *Osmanthus fragrans* (Mean = 2.96 ± 0.49 °C) and *Jacaranda* tree (Mean = 3.08 ± 0.46 °C).

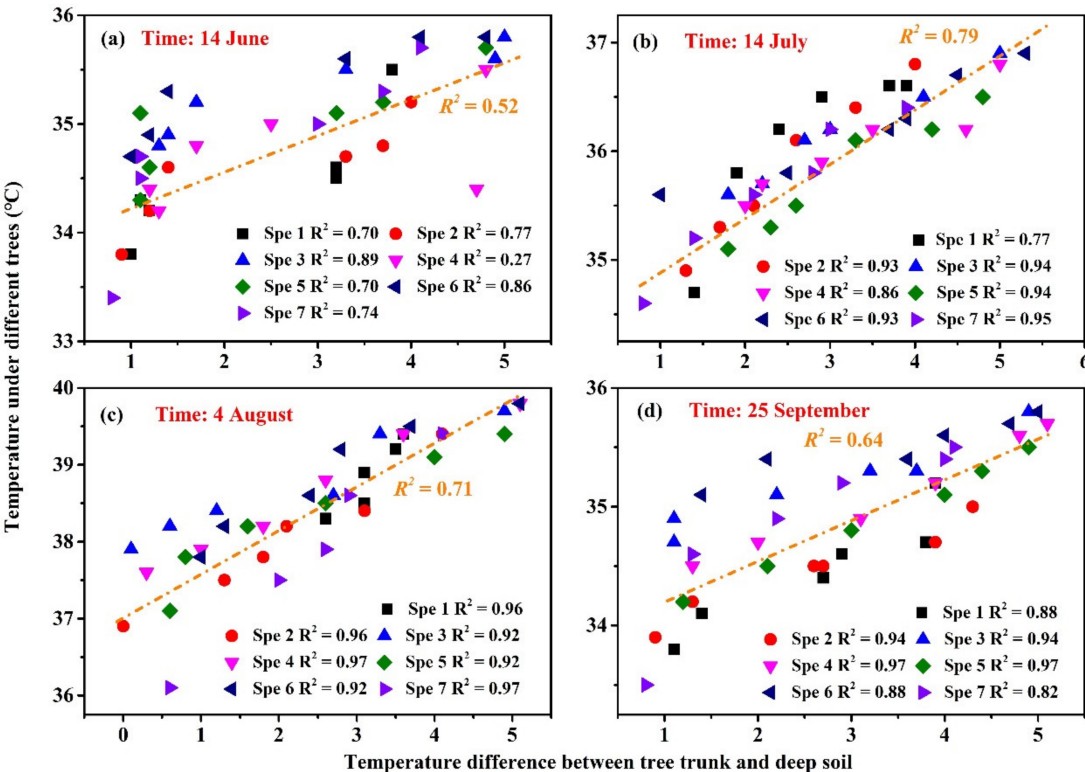

**Figure 4.** The relationship between temperature under trees and temperature difference between tree trunk and deep soil. Note that Spe 1–7 are *Ficus virens*, *Camphor* tree, *Ginkgo biloba*, *Osmanthus fragrans*, *Magnolia* tree, *Jacaranda* tree, and *Ficus macrocarpa*, respectively. (**a–d**) are for June, July, August, and September, respectively.

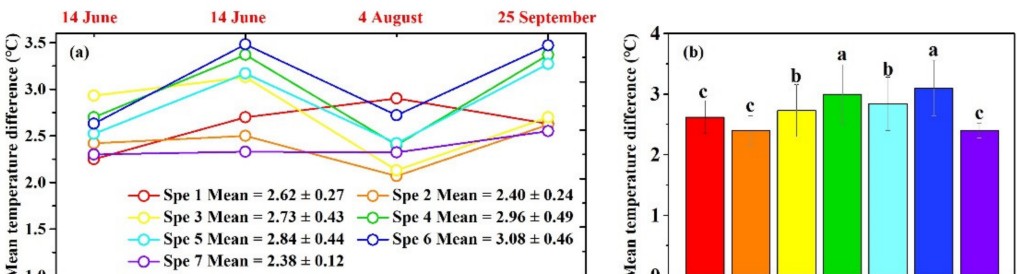

**Figure 5.** Significant difference analysis of temperature difference between tree trunk and deep soil for different species of trees. (**a**,**b**) is the significant difference analysis for mean temperature difference of different species. Note that Spe 1–7 are *Ficus virens*, *Camphor* tree, *Ginkgo biloba*, *Osmanthus fragrans*, *Magnolia* tree, *Jacaranda* tree, and *Ficus macrocarpa*, respectively.

### 3.3. Multiple Linear Regression for Control Factors

As shown in Figure 6, the trend of temperature difference in trees with different DBH sizes is obviously the opposite, with several controlling factors, such as canopy area, canopy height, LAI, and DBH. Canopy area, canopy height, and LAI of each species exhibited obvious differences among different DBH sizes. The highest DTa (Mean = $3.01 \pm 1.49$ °C) is for species 6 (*Jacaranda* tree), followed by species 4 (*Osmanthus fragrans*, Mean = $2.96 \pm 1.51$ °C) and species 5 (*Magnolia* tree, Mean = $2.84 \pm 1.45$ °C). However, the trend of the curve of control factors is not exactly the same. In detail, the highest temperature of the species is Spe 6 (*Jacaranda* tree), but the species with the lowest canopy area, canopy height, and DBH is Spe 4 (*Osmanthus fragrans*). In our study, multiple linear regression analysis was performed for multiple control factors as shown in Table 1. We found that canopy height is not the controlling factor for the temperature difference between the tree trunk and soil

deep, and the canopy area is colinear with LAI. Therefore, only DBH and LAI were selected to establish the multiple linear regression equation.

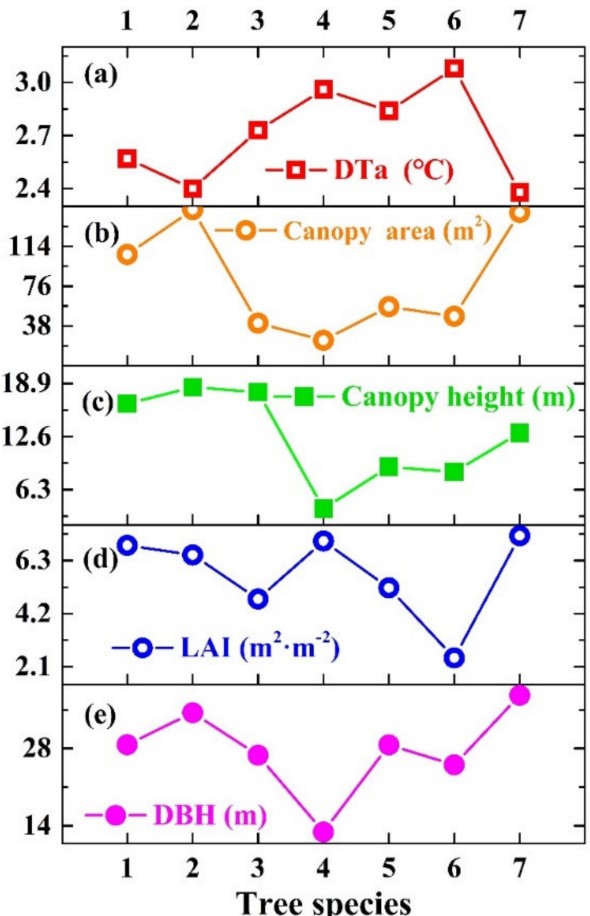

**Figure 6.** Comparison of control factor trend for different trees. (**a–e**) are temperature difference, canopy area, canopy height, LAI and DBH of different species, respectively. Note that 1–7 are *Ficus virens*, *Camphor* tree, *Ginkgo biloba*, *Osmanthus fragrans*, *Magnolia* tree, *Jacaranda* tree, and *Ficus macrocarpa*, respectively. DTa is temperature difference between tree trunk and deep soil.

**Table 1.** Multiple linear regression results between DTa and control factors. Note that DTa is temperature difference between tree trunk and deep soil, CH is canopy height, and CA is canopy area, respectively.

|  |  | Intercept | DBH | LAI | CH | CA | $R^2$ |
|---|---|---|---|---|---|---|---|
| Result 1 |  | 6.2943 | −0.0832 | −0.2561 | −0.0291 | 0.0068 | 0.6821 |
|  | *p*-value | 0.0000 | 0.0002 | 0.0021 | 0.3141 | 0.0650 |  |
| Result 2 |  | 6.2463 | −0.0926 | −0.2580 |  | 0.0063 | 0.6818 |
|  | *p*-value | 0.0000 | 0.0000 | 0.0019 |  | 0.0850 |  |
| Result 3 |  | 5.6671 | −0.0665 | −0.1942 |  |  | 0.6644 |
|  | *p*-value | 0.0000 | 0.0000 | 0.88 |  |  |  |

## 4. Discussion

The air temperature is positively correlated with DBH in our study. In detail, for all tree species, the air temperature under the trees with larger DBH is lower than that with smaller DBH, as shown in Figure 3a–d. In fact, DBH is only a superficial factor, but there are other factors that really affect temperature. In general, a larger DBH means a larger canopy for the same tree species [30]. On the one hand, large trees have a stronger ability to store water [31]. Understandably, plants are cooler than air, and larger plants mean a larger heat

transfer area. On the other hand, large trees have a stronger ability to evapotranspiration, which process consumes a lot of heat. Previous studies have demonstrated that taller trees have a great cooling effect due to evapotranspiration and shading [15]. Moreover, evapotranspiration causes the water originally stored in the plant body to dissipate into the air, and at the same time, absorb water from the deep soil to replenish the trunk and leaf [32,33]. A higher evapotranspiration rate means that water in your body is replaced more quickly, which means your body is cooler. This makes heat transfer more efficient and effective; that is, cooling the air around the plant more noticeable [34].

The temperature under trees and the temperature difference between the tree trunk and deep soil exhibited a linear positive correlation for all species as shown in Figure 4 in our study. This means that the closer the temperatures of the tree trunk and deep soil are to each other, the faster the water in the plant is replaced. Obviously, the giant trees with bigger DBH absorb deeper soil water more quickly. In this study, we used 60 cm as the depth representative of deep soil. In fact, the giant trees in this study are likely to absorb deeper soil water [32,33]. This also makes the trunk of these trees cooler than that of smaller trees. In terms of root distribution, the roots of large trees are significantly deeper [35–37], especially for trees with taproots well over 60 cm deep [35,38]. This is confirmed by root depth estimates using bottom-sounding radar.

In our study, when constructing multiple linear regression equations based on multiple control factors, only temperature was selected to evaluate the cooling effect. In fact, for each tree, the cooling effect should include not only the temperature, but also the area of cooling. In this study, the area of cooling was basically equal to the canopy area. Therefore, the canopy area, an important factor, was excluded when the stepwise regression equation was established. As for canopy height, it is not a key factor in the cooling effect between different species. Specifically, in this study, ginkgo and jacaranda trees, although tall, have relatively weak lateral branch growth, resulting in sparse leaf distribution, poor shading effect, weak transpiration capacity, slow water renewal rate in plants, and the weakest cooling effect. Therefore, canopy height is not a key factor in the cooling effect.

## 5. Conclusions

In this study, seven species of trees with the highest planting rate were selected in seven residential areas as research objects. By measuring the temperature under the trees and related control factors, the results conclude that: (1) Giant trees have the best cooling effect, i.e., the air temperature under the trees with larger DBH is lower than that with smaller DBH. The reason is that the high rate of water replenishment in plants increases giant trees' transpiration rate and cooling effect. (2) The closer the temperatures of the tree trunk and deep soil are to each other, the faster the water in the plant is replaced. The giant trees with bigger DBH often have larger crowns and deeper roots, as well as greater evapotranspiration; therefore, they often absorbed deeper soil water more quickly, which makes the trunk of these trees cooler than that of smaller trees. In terms of root distribution, the roots of large trees are significantly deeper. This study is of great practical significance to the selection of plants in the green belt of residential communities, and has great theoretical significance to understand the principle of the plant cooling effect.

**Author Contributions:** Conceptualization, R.Z. and Z.Z.; methodology, R.Z.; software, R.Z.; validation, R.Z. and Z.Z.; formal analysis, R.Z.; investigation, Z.Z.; resources, R.Z.; data curation, Z.Z.; writing—original draft preparation, Z.Z.; writing—review and editing, R.Z.; visualization, R.Z.; supervision, R.Z.; project administration, R.Z.; funding acquisition, R.Z. All authors have read and agreed to the published version of the manuscript.

**Funding:** This study was supported by the National Natural Science Foundation of China (42101036), the GDAS' Project of Science and Technology Development (2020GDASYL-20200103078) and the Natural Science Foundation of Guangdong Province (2020A1515111060), China Postdoctoral Science Foundation (2022M710504), Chongqing Postdoctoral Science Foundation (2021XM2031) and Fundamental Research Funds for the Central Universities (2022CDJXY-017).

**Acknowledgments:** This study was supported by the National Natural Science Foundation of China (42101036), the GDAS' Project of Science and Technology Development (2020GDASYL-20200103078) and the Natural Science Foundation of Guangdong Province (2020A1515111060), China Postdoctoral Science Foundation (2022M710504), Chongqing Postdoctoral Science Foundation (2021XM2031) and Fundamental Research Funds for the Central Universities (2022CDJXY-017). We thank the editors and anonymous reviewers for their valuable comments and suggestions for improving the manuscript.

**Conflicts of Interest:** We certify that we have participated sufficiently in the work to take public responsibility for the appropriateness of the experimental design and method, and the collection, analysis, and interpretation of the data. We have reviewed the final version of the manuscript and approved it for publication. This manuscript has not been published in whole or in part nor is it being considered for publication elsewhere.

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
