# Peer review of "Giant Trees Exhibited Great Cooling Effect in Residential Area Southwest of China"

_forests, doi:10.3390/f13091516_

Round 1

Reviewer 1 Report

The study presents measurements of air temperature under 7 different types of trees. While the topic is interestijng and of great importance to the field, the measurements are not designed properly. Information and measurement of the local climate are missing, which makes the findings not conclusive. Below are my comments for the authors to consider.

1. What are the solar radiation and wind conditions around these 7 types of trees over the measurment period?

2. To what extent you believe the local climate remains the same for different types of trees.

3. How was the air temperature measured? Did you manage to get rid of the influence of solar radiation on temperature sensors?

4. Would it be possible that your measurements were affected by shade from local buildings from trees to trees.

5. Violin plots may help you present most of the measurement data by giving most of important statistics. The fitting of data does not provide too much insight into the data.

6. Som closely related literatures are missing, just as an examples, "Urban heat island and its interaction with heatwaves: A review of studies on mesoscale."

7. Please improve the resoltuion/quality of all your images/plots.

Author Response

Author's Reply to the Review Report (Reviewer 1)

The study presents measurements of air temperature under 7 different types of trees. While the topic is interestijng and of great importance to the field, the measurements are not designed properly. Information and measurement of the local climate are missing, which makes the findings not conclusive. Below are my comments for the authors to consider.

  1. What are the solar radiation and wind conditions around these 7 types of trees over the measurment period?

Reply: All these 7 types of trees were near to each other (< 1 km). Therefore, solar radiation and wind conditions are same for them.

  1. To what extent you believe the local climate remains the same for different types of trees.

Reply: The local climate is almost total same for different types of trees, because the distant among them is very close (< 1 km).

  1. How was the air temperature measured? Did you manage to get rid of the influence of solar radiation on temperature sensors?

Reply: We have added the description about air temperature measure method as below:

“Air temperature sensors (10TH) were used to measure air temperature automatic continuously once every 5 minutes at 1.5 m height.” Please see the revision in line XXX. Because the solar radiation is affected by LAI and it is one of the reasons that air temperature is different among trees.

  1. Would it be possible that your measurements were affected by shade from local buildings from trees to trees.

Reply: We have considered the impact by shade from local buildings, therefore, when we select trees, we only select those can’t be shaded by buildings.

  1. Violin plots may help you present most of the measurement data by giving most of important statistics. The fitting of data does not provide too much insight into the data.

Reply: We admitted that Violin plots may present more results. But here, the fitting of data have provided what we wanted and we don’t have to explain superfluous results presented on violin plots.

  1. Som closely related literatures are missing, just as an examples, "Urban heat island and its interaction with heatwaves: A review of studies on mesoscale."

Reply: Done.

Kong, J. , Zhao, Y. , Carmeliet, J. , & Lei, C. . (2021). Urban heat island and its interaction with heatwaves: a review of studies on mesoscale. Sustainability, 13.

  1. Please improve the resoltuion/quality of all your images/plots.

Reply. Done.

Reviewer 2 Report

The manuscript is very interesting; however, there are small details that need to be addressed. For example, a little in the style to describe a part of the introduction in order to give it more clarity. In addition, in the methodology indicate the criteria for deciding the number of residential areas. Finally, there is some part of the conclusions that must verify if it can be concluded with the results. More details are shown in the attached PDF.

Author Response

Author's Reply to the Review Report (Reviewer 2)

The manuscript is very interesting; however, there are small details that need to be addressed. For example, a little in the style to describe a part of the introduction in order to give it more clarity. In addition, in the methodology indicate the criteria for deciding the number of residential areas. Finally, there is some part of the conclusions that must verify if it can be concluded with the results. More details are shown in the attached PDF.

  1. change the writing style... describe all the main results and then place the reference. MAIN IDEA AND RESULTS (CITE or CITES).

Reply: Done. We have rewrite this paragraph.

  1. This text looks a methodology. Make a broad idea; for example, Therefore, temperature under different tree species at residential communities were studied in ... in order to clarify the related control factors. The objectives of the paper are: 1)

Reply: Done.

  1. the picture is out of the margins. Please align it

Reply: Done.

  1. the text is out of the margins. Please align it

Reply: Done.

  1. I am not sure that you can conclude this with your results. Please rewrite

Reply: Done.

  1. some of this information has to be stated at the funding section

Reply: Done.

Round 2

Reviewer 1 Report

The manuscript has been improved to some extent. However, three points have to be addressed further before it could be accepted for publication in forest.

1. The title of the manuscript is very missleading and vague. What do you mean by using 'heat conduction'? Shading and transpirative cooling are the two major factors leading to cooling effect on air. High rate of water replensishment in plants should be related to high transpiratvie cooling. I can't see the point to attribute enhanced cooling effect to heat conduction. Also, the work is a field measurement, not a modeling nor simulation. Please reflect these points in the title of your manuscript.

2. Limitation of the field measurements. It's clear that some local climate conditions from one site to another were not measured. I understand measuring all local climate conditions could be challenging. However, these limiations should be noted down clearly in a separate section tilted ' Limitations of the field measurements'.

3. Literature. As I mentioned in my earlier review comments, there are a lot of recent studies concerning cooling effects of trees. Just as two more examples: 'Advancement in urban climate modelling at local scale: urban heat island mitigation and building cooling demand', and ‘Parametric study of the influence of environmental factors and tree properties on the transpirative cooling effect of trees'. Please do a more thorough literature review and discuss and include those most recent studies.

Author Response

The manuscript has been improved to some extent. However, three points have to be addressed further before it could be accepted for publication in forest.

  1. The title of the manuscript is very missleading and vague. What do you mean by using 'heat conduction'? Shading and transpirative cooling are the two major factors leading to cooling effect on air. High rate of water replensishment in plants should be related to high transpiratvie cooling. I can't see the point to attribute enhanced cooling effect to heat conduction. Also, the work is a field measurement, not a modeling nor simulation. Please reflect these points in the title of your manuscript.

Reply:We have changed the title to “Giant trees exhibited great cooling effect through heat conduction in residential area”. Of course, Shading and transportive cooling are the two major factors leading to cooling effect on air, and this study also admitted and described that in text. However, our study concentrated on heat conduction for trees, it is the reason for the emphasis on heat transfer in the title.

  1. Limitation of the field measurements. It's clear that some local climate conditions from one site to another were not measured. I understand measuring all local climate conditions could be challenging. However, these limiations should be noted down clearly in a separate section tilted ' Limitations of the field measurements'.

Reply:I'm a little confused about this question. The data such as air temperature, soil temperature and the temperature of trees’ trunk were measured by sensors for all 7 residential areas and we described that in text.

  1. Literature. As I mentioned in my earlier review comments, there are a lot of recent studies concerning cooling effects of trees. Just as two more examples: 'Advancement in urban climate modelling at local scale: urban heat island mitigation and building cooling demand', and ‘Parametric study of the influence of environmental factors and tree properties on the transpirative cooling effect of trees'. Please do a more thorough literature review and discuss and include those most recent studies.

Reply:We have added some references based on earlier review comments. We also have read the two papers 'Advancement in urban climate modelling at local scale: urban heat island mitigation and building cooling demand', and ‘Parametric study of the influence of environmental factors and tree properties on the transportive cooling effect of trees'. However, our study was concentrating on heat conduction for cooling effect and the two papers did not mention that. That’s why we did not cite these two articles and similar articles.
